# Prevalence of Aflatoxins in Selected Dry Fruits, Impact of Storage Conditions on Contamination Levels and Associated Health Risks on Pakistani Consumers

**DOI:** 10.3390/ijerph19063404

**Published:** 2022-03-14

**Authors:** Iqra Naeem, Amir Ismail, Awais Ur Rehman, Zubair Ismail, Shehzadi Saima, Ambreen Naz, Asim Faraz, Carlos Augusto Fernandes de Oliveira, Noreddine Benkerroum, Muhammad Zahid Aslam, Rashida Aslam

**Affiliations:** 1Faculty of Agricultural Sciences and Technology, Institute of Food Science and Nutrition, Bahauddin Zakariya University, Multan 60000, Pakistan; niqra5771@gmail.com (I.N.); awaisfst43@gmail.com (A.U.R.); zubairismail001@gmail.com (Z.I.); 2Faculty of Sciences, Institute of Pure and Applied Biology, Bahauddin Zakariya University, Multan 60000, Pakistan; shehzadi.saima@bzu.edu.pk; 3Department of Food Science and Technology, MNS University of Agriculture, Multan 60000, Pakistan; amber1912@yahoo.com; 4Department of Livestock & Poultry Production, Faculty of Veterinary Sciences, Bahauddin Zakariya University, Multan 60000, Pakistan; drasimfaraz@bzu.edu.pk; 5Department of Food Engineering, School of Animal Science and Food Engineering, University of São Paulo, Pirassununga 13635-900, Brazil; carlosaf@usp.br; 6Canadian Food Inspection Agency, 93 Mount Edward Rd Charlottetown, Charlottetown, PE C1A 5T1, Canada; 7Cotton Research Station (CRS), Bahawalpur 63100, Pakistan; m.zahid1342@gmail.com; 8Cytogenetics Section, Central Cotton Research Institute, Multan 60000, Pakistan; rashidaaslam89@gmail.com

**Keywords:** dry fruits, dates, walnuts, pistachios, aflatoxin, risk assessment, margin of exposure (MoE), storage, open air, cold storage

## Abstract

Dry fruits and nuts are nutritious foods with several health-promoting properties. However, they are prone to contamination with aflatoxins at all stages of production and storage. The present study aimed to determine the natural occurrence of aflatoxin B1 (AFB1), aflatoxin B2 (AFB2), aflatoxin G1 (AFG1), aflatoxin G2 (AFG2), and total aflatoxins (AFT) in dates, pistachios, and walnuts collected from four districts of South Punjab (Pakistan), and to assess the associated health risks as estimated by dietary exposure and the Margin of Exposure (MoE) determinations. The contents of AFB1 and AFT in these food products were monitored during storage under three different conditions (open-air, hermetically closed jars, and refrigeration at 4 °C) to determine the most efficient conditions in preventing aflatoxin accumulation. HPLC-fluorescence analysis of 60 samples of these products for aflatoxin contamination showed that 52 (86.7%) samples were contaminated at different levels, with a maximum of 24.2 ng/g. The overall (all samples) mean concentrations of AFB1, AFB2, AFG1, AFG2, and AFT were 3.39 ± 2.96, 1.39 ± 1.68, 1.63 ± 1.48. 1.12 ± 1.23, and 7.54 ± 6.68, respectively. The Estimated Daily Intake (EDI) and MoE of aflatoxins through the consumption of the products ranged from 0.06 ng/kg bw/day to 2.0 ng/kg bw/day and from 84.84 to 2857.13, respectively, indicating that consumers are at high health risk. Significant differences were recorded between aflatoxin levels in the samples stored under different storage conditions, with storage under refrigeration (4 °C) being the most effective in controlling aflatoxin accumulation, although storage in closed jars was also efficient and offers a more flexible alternative to retailers. The findings of the study urge official authorities of Pakistan to implement appropriate regulatory and control measures and surveillance program to alleviate the potential public health risks associated with the consumption of dry fruits and nuts in the scope of their increased consumption.

## 1. Introduction

Aflatoxins are difumarocoumarin derivatives synthesized through the polyketide pathway by several fungal species, mainly Aspergillus flavus, A. parasiticus, and A. nomius, widely distributed contaminants of a variety of agri-food commodities [1]. More than twenty different types of aflatoxins have been identified so far, among which B-group (aflatoxins B_1_ and B_2_) and G-group (aflatoxins G_1_ and G_2_) are the most frequent and toxic ones [2]. Aflatoxin B_1_ (AFB_1_) and natural mixtures of aflatoxins are classified by the International Agency for Research on Cancer (IARC) as human carcinogens of group 1 [3]. Moreover, aflatoxins have been associated with acute and chronic health conditions, including hepatotoxicity, teratogenicity, genotoxicity, retarded growth, immunotoxicity, and neurotoxicity in both humans and animals [4].

The consumption of nuts and dried fruits is being encouraged around the world, not only for their high nutritional quality and palatability [5,6], but also because they constitute a valuable source of bioactive substances with established health-promoting properties [7,8]. Indeed, there is a growing body of evidence that the consumption of nuts and dried fruits is inversely related to the risk of all-cause cardiovascular disease (CVD) and total cancer mortality [7,8]. This has been substantiated by numerous in vitro, in vivo, clinical, and epidemiological studies demonstrating that the consumption of nuts and dried fruits improves or modulates the underlaying biological parameters of most diseases to prevent their onset or alleviate their endpoints. Such studies have demonstrated the positive effects of nut and dry fruit consumption on the blood lipid profile, insulin and glucose homeostasis, body weight gain, bone health, cardio-metabolic health, appetite and satiety, oxidative stress, and gastrointestinal physiology via the modulation of its microbiome [5,7,8]. Therefore, food scientists, nutritionists, as well as international and regional food authority organs, highly recommend promoting the consumption of nuts and dried fruits individually or together in enhanced disease-preventive and high quality diets, owing to their complementary nutritional and health attributes [5,7,9]. The US Food and Drug Administration (FDA) and the European food Safety Authority (EFSA) recommend the consumption of 42.5 g and 30 g of nuts per day, respectively [10,11]. Additionally, consistent with the dietary guidelines for dried fruit consumption in the US (112 g/day) and England (150 g/day), a daily consumption of 100 g to 150 g of dried fruits (equivalent of 200 and 300 g of fresh fruits) was reported to be optimal for the prevention of most diseases, thereby expanding life expectancy [12].

On the other hand, dried fruits and nuts are particularly susceptible to fungal contamination and aflatoxin accumulation during the pre- and post-harvest production processes [13,14]. The levels of contamination are particularly high with blemished harvesting, inadequate drying, and faulty storage conditions [15]. Pistachios and walnuts are among the most commonly consumed tree nuts in the world, and they are exposed to moderate to high risks of mold attack and aflatoxin production, as they are usually grown and harvested under environmental conditions that favor aflatoxigenic fungal growth and the production of aflatoxins, which further accumulate during storage [16,17]. To safeguard consumers’ health, aflatoxins are regulated in nuts and dry fruits by several regulatory authorities from different parts of the world, with a Maximum Tolerable Limit (MTL) of 2 ng/g for AFB_1_ and 4 ng/g for total aflatoxins (AFT) in the European Union [18]; 5 ng/g for AFB_1_ and 10 ng/g for AFT in Australia, New Zealand, Canada, Turkey and United States; and 5 ng/g for AFB_1_ and 15 ng/kg for AFT in Iran [19,20,21]. However, most developing countries adopt the relaxed figures of 8 ng/g for AFB_1_ and 20 ng/g for AFT [20], while others, including Pakistan, do not regulate aflatoxins in dry fruits. Nonetheless, the recently implemented MTL of 20 ng/g for AFT in Pakistan for all food articles applies to dried fruits and nuts [22].

Walnuts and pistachios are the second and third most produced tree nuts worldwide, with a total production of 1022.03 thousand MTs and 1008.8 thousand MTs, respectively, in the year 2020–2021 [23]. They also ranked third and fifth in 2018 among the most consumed nuts, with consumption quantities of 1188.69 thousand MTs and 669.86 thousand MTs, respectively [23]. In the year 2019, Pakistan ranked 6th and 50th in the world for date and nut production, respectively, with total production of 455.99 thousand metric tons (MTs) of dates and 2889.38 MTs of nuts [24,25]. Among the dried fruits produced in Pakistan, the government has explicitly prioritized dates as a subsistence crop in vast desert areas [26].

Controlling mold growth and aflatoxin production in stored commodities is a major challenge facing food safety and public health, particularly in the developing world. Food storage systems are intended to limit the growth of spoilage microorganisms by providing mechanical barriers or unfavorable environmental conditions [27]. Various studies have reported on the prevalence of aflatoxins in dry fruits from different parts of the world [28,29,30,31,32]. However, limited information is available regarding the effects of storage conditions on the levels of aflatoxins in these food products. Although studies have been carried out to evaluate aflatoxin contamination in dry fruits in Pakistan [33,34,35], no dietary exposure assessment, to our knowledge, has been conducted to evaluate the potential risk associated with the consumption of aflatoxin-contaminated dates and nuts.

The present study aimed to determine the natural occurrence of aflatoxins in selected dry fruits (dates, pistachios, and walnuts) from four districts of South Punjab, Pakistan, and to estimate the dietary exposure to aflatoxins and the associated health risks posed to the population of the region through the consumption of these commodities. The influence of simulated storage conditions on aflatoxins levels in these foods was also evaluated to determine the most appropriate storage means that would help preventing or reducing aflatoxin accumulation.

## 2. Materials and Methods

### 2.1. Sampling

A total of 60 samples composed of 5 samples each of dates, pistachios, and walnuts collected from the local market of each of the four districts (Bahawalpur, Dera Ghazi Khan, Multan, and Rahim Yar Khan) of South Punjab, Pakistan, was used in this study. These major districts of South Punjab—Pakistan, are representative of South Punjab in terms of population and dietary habits, and they have not been studied for this purpose before. Meanwhile, the results of this study would be complementary to those previously reported for the northern part of the province [36,37] to provide a larger understanding of the status of dry fruit contamination with aflatoxins. The South Punjab climate is typically arid with less rainfall than the semi-arid northern part of the province [38], which was expected to influence the extent of contamination with aflatoxins in the studied products. The samples were collected during the period spanning from June 2020 to January 2021. A 2 kg sample of each commodity was drawn from bulk containers, placed into a zip-lock bag, and stored at 4 °C in a refrigerator until analysis. The times of exposure for sale in the stores before sampling were not known.

### 2.2. Aflatoxins Extraction and Immuno-Affinity Cleanup

The extraction and clean-up method described previously [39] was used in this study. An aliquot of 25 g of finely ground sample was mixed thoroughly with a 100 mL solution of methanol and water (60/40 ratio) using an orbital shaker (Thermoscientific MaxQ 4000, Waltham, MA, USA) at 200 rpm. The extracted solutions were filtered using Whatman filter paper No. 42 and the resulting 4 mL filtrates were diluted up to 20 mL by adding 50 mM Phosphate Buffered Saline (PBS) at pH 7.4. The diluted extracts were passed through immunoaffinity columns (Eurofins, Siegen, Germany) for clean-up as recommended by the manufacturer.

### 2.3. Derivatization

The derivatization of aflatoxins was performed according to the official method of AOAC, August 2005 [40]. Briefly, about 200 µL of hexane and 50 µL of trifluoroacetic acid (TFA) were added to glass vials containing dried extracts of aflatoxins. The glass vials were then tightly closed and left to react in the dark for approximately 5 min. A solution of double distilled water and acetonitrile (9:1) was added in the vials to make up the solution to the final volume of 1.95 mL, and this was vortex for 1 min to separate layers. The lower aqueous layer containing aflatoxins was separated with syringe filters (0.45 µm) to eliminate putative small particles, reduce the matrix effects and protect the HPLC system before proceeding to chromatographic analysis.

### 2.4. Chromatographic Analysis

A High-Performance Liquid Chromatography (HPLC) system S 500 routine series equipped with an S1125 isocratic pump (Sykam, Eresing, Germany) and coupled to a fluorescence detector as described previously [39] with some adjustments was used to quantify aflatoxins. The mobile phase was prepared using water/methanol/acetonitrile (65/25/15, *v*/*v*/*v*) at a flow rate of 1 mL/min. The column oven temperature was adjusted to 37 °C. The excitation and emission wavelengths of the fluorescence detector (Sykam, RF-20A) were set at 365 nm and 440 nm, respectively. A reverse phase C18 column (Welch Material, Inc., Austin, TX, USA) (4.6 × 250 mm) was used as the stationary phase. Aflatoxin standards—AFB_1_ (0.5 µg/mL), AFB_2_ (0.25 µg/mL), AFG_1_ (0.5 µg/mL), and AFG_2_ (0.25 µg/mL)—of HPLC-grade purity (≥98%) were purchased from Sigma-Aldrich (St. Louis, MO, USA). Each standard and sample were injected at a volume of 20 µL with a run time of 22 min. The retention time for AFB_1_ was 5.09 min, 8.78 min for AFB_2_, 4.28 min for AFG_1_, and 6.66 min for AFG_2_.

### 2.5. Method Validation

The analytical method was validated in terms of linearity, sensitivity, and accuracy. The linearity was assessed using five-point calibration curves of the analytes at the concentrations of 1, 5, 10, 25, and 50 ng/mL. The accuracy of the aflatoxin measurements was analyzed by determining their recoveries. Aflatoxin-free samples of dates, pistachios, and walnuts were spiked with 30, 60, and 90 ng/g of AFT (sum of AFB_1_, AFB_2_, AFG_1_, and AFG_2_) at the ratio of 1.0:0.5:1.0:0.5. Recoveries were calculated by comparing the aflatoxin concentrations in the spiked samples with the levels in the reference standard solutions. The percent recoveries of aflatoxins were calculated using Equation (1).
(1)$$\mathrm{Recovery}\;(\%)=\frac{\mathrm{measured}\;\mathrm{concentrations}}{\mathrm{spiked}\;\mathrm{concentrations}}\times 100$$

The sensitivity of the method was evaluated in terms of limit of detection (LOD) and limit of quantification (LOQ) computed according to the method of Golge et al. [16]. Blank samples were spiked with aflatoxins and measured in 12 independent replicates. The LOD and LOQ values were calculated according to Equations (2) and (3);
(2)LOD=3×SD
(3)LOQ=10×SD
where SD is the standard deviation of the sample determinations. All the experiments were performed in triplicates and the reliability of the results was confirmed by analyzing each sample thrice.

### 2.6. Aflatoxins Exposure and Health Risk Assessment

A cross-sectional survey was conducted from November 2020 to February 2021 to estimate the average daily consumption of dates, pistachios, and walnuts by male (n = 200) and female (n = 200) adult residents of South Punjab, Pakistan. A structured questionnaire developed for the purpose of this study and administered by trained interviewers was used to procure the information regarding socio-demographics, purchases, and storage patterns of dates and nuts, along with other information as appropriate. The individuals were asked to provide information about the number of servings of studied items consumed each time, either “per day”, “per week”, “per month”, “per year”, or “never’’, and their storage-related preferences, i.e., open air storage, storage in closed containers/glass jars, or cold storage at both household level and at the time of purchase based on the past one-year routine. The weights of participants were determined at the time of completing the questionnaire by using the calibrated electronic balance Westpoint Weight Scale, WS-7009 (Gaba Electronics, Karachi, Pakistan).

### 2.7. Exposure Assessment

The consumption rate (CR) in g/person/day of dates, pistachios and walnuts was calculated by multiplying the consumption frequency of each of the studied food items with the total number of servings and weight of food (kg) per serving. The estimated daily intake (EDI) of the studied items was calculated according to Equation (4) as defined by the FAO/WHO [41];
EDI = C × CR/W_AB_(4)
where EDI is the estimated daily intake expressed in ng/kg bw/day; C is the mean concentration of aflatoxins in ng/g determined in each food item; CR is the daily consumption rate of the food product expressed in g/person/day; W_AB_ is the average body weight expressed in kg of the surveyed participants. The EDI values were calculated by computation using Microsoft Excel 2013 version (Microsoft Corporation, Redmond, Washington, DC, USA).

### 2.8. Health Risk Characterization

The health risk for cancer development associated with the consumption of dry fruits contaminated with aflatoxins was estimated using the Margin of Exposure (MoE) approach of the European Food Safety Authority (EFSA) [18]. A Benchmark Dose Lower confidence Limit (BMDL) of 170 ng/kg bw/day obtained from animal modeling studies [18,27] was considered. The MoE value for the studied subjects was calculated using Equation (5);
MoE = BMDL_10_/Exposure(5)

MoE < 10,000 indicates a greater public health concern related with exposure to genotoxic carcinogens [42,43]. The lower the MoE, the higher the risk associated with the consumption of the contaminated food product.

The mean EDI values were compared with the recently suggested Tolerable Daily Intake (TDI) values of 0.017 ng/g/kg bw/day and 0.82 ng/g/kg bw/day for the non-carcinogenic adverse health effects, such as the immunotoxicity, of aflatoxins [44].

### 2.9. Effect of Storage Conditions on the Levels of Aflatoxins

To evaluate the impact of storage conditions on the evolution of aflatoxin concentration as a function of the storage time, a bag of 5 kg of the best quality of each of the studied items was purchased from the local market. The best quality of the samples was judged visually as described by Tangendjaja et al. [45] with an emphasis on mold growth, and a subsample of 3 kg was drawn after thoroughly mixing the samples. From the collected stock of dried fruits (3 kg), three lots of 1 kg each were randomly prepared and a total of 9 samples (25 g each) were randomly collected from each lot for the determination of aflatoxin concentrations at day zero. The lots were then subjected to storage under conditions simulated in the laboratory of the Institute of Food Science & Nutrition, Bahauddin Zakariya University Multan, Pakistan, for periods of 60 and 90 days. The three storage conditions most used for the conditioning and storage of foods in retail stores were selected for this study: (a) open-air storage, as is normally practiced in the local market from which our samples were taken, (b) hermetic storage to simulate unfavorable anaerobic conditions for mold growth, as described by Duman [46], and (c) cold storage under unfavorable temperatures for mold growth and/or toxigenesis. The samples to be stored in the open air were piled on jute sheets placed on the floor and were kept at room temperature. The hermetic storage was done by placing the samples in glass jars of 1.0 kg capacity, which were kept at room temperature. The lids of the glass jars were opened 6 times in a day for 3–5 min, as is commonly practiced by local retailers in Pakistan, to assess the effect of periodic opening on the level of aflatoxins compared with the control (aflatoxin-free samples). The samples to be stored under cold storage were maintained at a temperature of 4 °C and a relative humidity (rH) of 65%. At the end of each storage period, samples from the lots stored under the three different storage conditions were analyzed for aflatoxin concentrations. All of the treatments were performed in triplicates.

### 2.10. Statistical Analysis

All the analyses were performed in triplicates. The obtained results were statistically evaluated using Statistix 8.1 software (Statistix 8.1, Tallahassee, FL, USA). The difference in aflatoxin concentrations at the significance level of 5% was evaluated using Analysis of Variance (ANOVA). For calculations of the mean values and standard deviations, Microsoft Excel 2013 version was used.

## 3. Results and Discussion

### 3.1. Aflatoxins Occurrence in Dates, Walnuts, and Pistachios

The results of the HPLC-fluorescence technique validation experiments suggest its reliability for the analyses carried out in the present study. A good linearity was found in the relationship between the analyte concentrations with coefficients of correlations greater than 95% (R^2^  > 0.9995). Additionally, the method proved accurate and sensitive, as substantiated by the high recovery percentages and the low limits of detection (LOD) and limits of quantification (LOQ). Table 1 shows that the recovery values of the analyzed types of aflatoxins ranged between 85.7% and 96.4%. Recovery rates varying from 68.7% to 104.8% were reported in similar studies [47,48,49,50,51], and The Joint Expert Committee of Food Additives and Contaminants (JECFA) of the World Health Organization (WHO) and the Food and Agricultural Organization (FAO) of the United Nations validated the methods of aflatoxin quantification with recovery rates higher than 70% [41,52]. The LOD values were 0.07 ng/g for AFB_1_ and AFG_1_ and 0.03 ng/g for AFB_2_ and AFG_2_, while the LOQ values were 0.21 ng/g for AFB_1_ and AFG_1_ and 0.09 ng/g for AFB_2_ and AFG_2_. These data suggest that the sensitivity of the method used in this study is in the range of acceptable sensitivities of the methods most commonly used worldwide, with LOD values ranging between 0.1 and 0.4 ng/g for AFB_1_ and 0.2 and 0.4 ng/g for AFT, and LOQ values ranging between 0.05 and 0.2 ng/g for AFB_1_ and 0.1 and 1.0 ng/g for AFT [41]. However, with the continuous advances in HPLC analytical methods, the LOD and LOQ values for the determination of aflatoxin concentrations now typically fall within the interval of 0.001–0.20 ng/g [53], which also includes the values obtained in this study.

The results for aflatoxin concentrations in dates, pistachios, and walnuts collected from four districts of South Punjab (Pakistan) are summarized in Table 2. Among the analyzed aflatoxins, AFB_1_ was the most frequently detected, with an overall average prevalence of 86.7%, followed by AFG_1_ (81.7%), AFG_2_ (60%), and AFB_2_ (58.3%). Statistical analysis revealed significant differences (*p* < 0.05) in the levels of aflatoxins among dates, pistachios, and walnuts. No significant difference (*p* > 0.05) was observed between the aflatoxin concentrations in samples of the same product collected from the four studied districts of Punjab, Pakistan, except for dates from Multan, whose AFT concentration was significantly different (*p* < 0.05) from those collected from the other districts (Table 2). The concentrations of AFT in the analyzed samples ranged between <LOD and 24.2 ng/g, with an average value of 7.54 ng/g. Individual aflatoxin concentrations, regardless of the district, were in the range of <LOD-9.99 ng/g, <LOD-4.85 ng/g, <LOD-5.63 ng/g, and <LOD-4.63 ng/g, for AFB_1_, AFB_2_, AFG_1_, and AFG_2_, respectively, with the corresponding average values of 3.39 ng/g, 1.39 ng/g, 1.63 ng/g, and 1.12 ng/g (Table 2). Regarding the magnitude of contamination of the studied commodities, these ranked in the order of walnuts > pistachios > dates, according to their respective average concentrations of 12.76 ng/g, 7.13 ng/g, and 2.73 ng/g for AFT and 5.52 ng/g, 3.32 ng/g, and 1.33 ng/g for AFB_1_ (Table 3). The higher levels of aflatoxin contamination in walnuts compared with pistachios has been reported in other studies from different countries [50,54,55]. Conversely, a higher contamination of pistachios than walnuts and dates was also reported earlier in Pakistan, where the respective mean concentrations in pistachios, walnuts, and dates were 6.47 ng/g, 4.80 ng/g, and 4.50 ng/g for AFB_1_, and 7.53 ng/g, 5.43 ng/g, and 6.32 ng/g for AFT [37]. In the samples analyzed in this study, the prevalence of AFT varied between 20% and 100%, depending on the food item and the district considered (Table 2). The results of Table 2 also show that the highest mean concentration of AFT (13.73 ng/g) was found in walnuts from Multan, while the lowest mean concentration (1.47 ng/g) was recorded in dates from the same district, i.e., Multan. Compared with the EU’s most stringent regulations on aflatoxins in nuts and dried fruits intended for direct human consumption or use as food ingredients [18], the maximum tolerable limit (MTL) of AFB_1_ (2 ng/g) and the MTL of AFT (4 ng/g) exceeded 60% (n = 36) and 58.3% (n = 35), respectively, of the analyzed samples (Table 4). However, only 10% (n = 6) and 5% (n = 3) of the samples exceeded the relaxed and most widely adopted MTLs of 8 ng/g for AFB_1_ and 20 ng/g for AFT, respectively (Table 4). The occurrence of aflatoxins in dates, pistachios, and walnuts from different parts of the world is well documented, with highly variable prevalence and levels of contamination, depending on the country or region of production and/or marketing. In a survey on the aflatoxin contamination of different dried fruits and nuts marketed in Northern Punjab and Khyber Pakhtunkhwa province of Pakistan, Masood et al. [37] showed that the prevalence of aflatoxins was 60% (9 out of 15) for dates, 45% (9 out of 20) for walnuts, and 70% (14 out of 20) for pistachios. Higher mean prevalence values were found in the samples collected from the four districts studied herein, which ranged between 58.3% and 86.7%, with the individual values (per product) varying between 20% and 100% (Table 2). As regards the contamination levels, discrepancies can be noted between our results and those reported by the latter authors [37], depending on the product. While both studies found similar contamination levels of pistachios with AFT, lower concentrations of AFB_1_ (range of 2.72–4.05 ng/g) (Table 2) were detected in the samples analyzed in this study. Additionally, compared to the results of the study conducted by Masood et al. [37] on the contamination levels of dates, our results show lower levels for AFB_1,_ but somewhat higher levels for AFT (7.54 ng/g vs. 6.32 ng/g). Moreover, higher concentrations of AFB_1_ and AFT were found in the walnut samples analyzed herein compared with those reported by the latter authors. In another study on the occurrence of aflatoxins in dried fruits and nuts collected from different regions of Pakistan, including North Punjab [36], the contamination levels of pistachios with AFT were in accordance with our findings, although we found slightly higher contents of AFB_1_ and AFT in dates and pistachios. Despite the moderate discrepancies between our results and those of the above-mentioned studies, they concur to suggest that the three studied commodities marketed in Pakistan are frequently contaminated with aflatoxins at levels that may raise public health concerns. The results suggest also that the aflatoxin contamination patterns of dry fruits and nuts marketed in both North and South Punjab, Pakistan, are similar, probably due to the same origins of the products and the use of the same marketing chain. High variations in the levels and frequency of dry fruits and nut contamination with aflatoxins, depending on the country, the region, and various other production factors and regulatory measures, are well-documented. For comparison purposes, Table 3 summarizes the results of studies conducted in different countries on the same food items analyzed in this study. The table shows that the aflatoxin contamination levels of pistachios and walnuts marketed in some countries are significantly higher than in ours, regardless of the levels of development. For example, despite the strict implementation of EU regulatory measures and the practices of modern agricultural countries in reducing the aflatoxin contamination of agri-foods, abnormally high levels of AFB_1_ and AFT were detected in pistachios marketed in Spain [56] and Italy [57]. This could be attributed to the origin of the samples, as was suggested by Diella et al. [57], who mentioned that 95% of the highly contaminated pistachio samples marketed in Italy were imported from countries of Eastern Asia, known to be an endemic region for aflatoxins [15]. Moreover, outstandingly high concentrations of AFB_1_, reaching 1430 ng/g and 2500 ng/g in pistachios and walnuts, respectively, were recorded in Morocco, with average values of 158 and 360 ng/g [54] (Table 3). Conflicting results were reported on the level of contamination of nuts in Saudi Arabia, depending on the product and the city. A study conducted on pistachios and walnuts collected from local markets in different cities of the country showed that pistachios were highly (up to 411 ng/g of AFB_1_ and 422 ng/g of AFT) contaminated, while walnuts were less heavily, although more frequently, contaminated with AFB_1_ and AFT [50]; the highest values reached were 17.4 for AFB_1_ and 36.6 for AFT (Table 3). Conversely, another study on the contamination of pistachios and walnuts marketed in Jeddah of the same country [58] found significantly lower concentrations, which were not considered to raise serious health concerns for consumers (Table 3). A similar situation could be noted in Turkey, where pistachios were shown to be highly contaminated with AFB_1_ and AFT [59], while another study demonstrated a moderate contamination of walnuts [60], reflecting inconsistencies in the supply of retailers and the marketing chain in the country. As can be seen from Table 3, moderate and yet significant concentrations of aflatoxins in dates, pistachios, and walnuts have been reported in different countries around the world, including Iran [61], Algeria [62], and Turkey [59].

Among dried fruits, dates have received the least research attention from the perspective of their contamination with aflatoxins. The available data suggest that they are less contaminated with aflatoxins compared with other dried fruits and nuts [63,64], as shown in Table 3. This is in agreement with our results, wherein the lowest concentrations of aflatoxins were found in the date samples collected from the four studied districts (Table 2). Similarly, a Tunisian study revealed that 46% (22 out of 48) of date samples were contaminated with AFT at levels below 2.2 ng/g [65]. Moreover, the Canadian Food Inspection Agency (CFIA) conducted a survey on figs and dates imported into Canada in the years 2009–2010 for their contamination with aflatoxins. The survey, which included 49 samples of dates imported from the USA, Tunisia, and Iran to provide a baseline that can be used for the official control of nut importations, revealed the absence of detectable levels of aflatoxins in all the samples [64]. However, relatively high levels of aflatoxins in dates were occasionally reported. For example, dates from Iran were shown to contain AFB_1_ and AFT at levels ranging from 0.6 to 6.6 ng/g and from 0.9 to 8.1 ng/g, respectively, with the respective average values of 2.1 and 2.6 ng/g [28]. Additionally, 8.33% (2 out of 24) samples of dates from Egypt were reported to be contaminated with concentrations as high as 300 ng/g and 390 ng/g [66], but these were shown to be free from aflatoxins in another study [67]. Nonetheless, according to Ozer et al. [63], the aflatoxin contamination of dates can be significantly reduced by sorting them to remove decayed or damaged dates.

**Table 3 ijerph-19-03404-t003:** Comparison between the contamination levels (ng/g) of pistachios, walnuts, and dates with aflatoxin B1 (AFB_1_) and total aflatoxins (AFT) obtained in the present study with those of similar studies conducted in North Punjab, Pakistan, and other countries around the world.

Country (Region or City)	Pistachios						Walnuts						Dates						Reference
	AFB_1_			AFT			AFB_1_			AFT			AFB_1_			AFT			
	Max	Mean	%*P*	Max	Mean	%*P*	Max	Mean	%*P*	Max	Mean	%*P*	Max	Mean	%*P*	Max	Mean	%*P*	
Pakistan (South Pounjab)	6.73	3.32	95	7.90	7.13	95	9.99	5.52	95	13.73	12.76	95	4.34	1.33	75	5.13	2.73	75	This study
Pakistan (North Pujab)	-	5.80	50	20.70	7.10	50	-	-	-	-	-	-	-	4.80	40	10.20	5.30	40	[36]
Pakistan (North Pujab)	13.67	6.47	70	21.50	7.53	70	11.50	4.80	45	15.78	5.43	45	9.80	4.50	60	18.79	6.32	60	[37]
Spain	1037.3	21.4	10	1134.5	34.5	10	-	-	-	-	-	-	-	-	-	-	-	-	[56]
Italy	354.5	53.3	50	387.3	70.5	50	-	-	-	-	-	-	-	-	-	-	-	-	[57]
India	<LOD	-	-	186.6 *	-	-	-	-	-	-	-	-	-	-	-	-	-	-	[68]
Morocco	1430	158	45	1450	163	45	2500	360	30	4320	730	30	-	-	-	-	-	-	[54]
Saudi arabia (Mekka)	411	-	-	422	16.6	34	17.4	-	-	36.6	3.4	50	-	-	-	-	-	-	[50]
Saudi arabia (Jeddah)	4.28	-	45.5	7.13	1.86	63.6	3.85	-	66.7	4.39	1.16	75	-	-	-	-	-	-	[58]
Iran	-	-	-	-	-	-	2.08	0.48	2.3	38.1	14.4	90.7	-	-	-	-	-	-	[55]
Iran	-	-	-	-	-	-	-	-		-	-		6.0	2.1	40.9	8.1	2.6	40.9	[28]
Iran	-	-	-	5.8	1.5	-	-	-	-	3.6	1.8	-	-	-	-	-	-	-	[61]
Algeria	8.72	4.45	87.5	13.45	6.70	87.5	6.34	3.42	75	8.76	4.90	75	-	-	-	-	-	-	[62]
Turkey	-	-	-	-	-	-	5.06	0.86	20	10.3	1.68	64	-	-	-	-	-	-	[60]
Turkey	368	4.55	14.6	385	4.95	14.6	-	-	-	-	-	-	-	-	-	-	-	-	[59]
Tunisia	-	-	-	-	-	-	-	-	-	-	-	-	ND	-	-	3.04	-	46	[65]

-: Not available; ND: not detected (<limit of detection; LOD); * one sample.

**Table 4 ijerph-19-03404-t004:** Prevalence of AFB_1_ and AFT in dates, pistachios, and walnuts and percentage of the samples exceeding the maximum tolerable limits (MTL) set by the European Union [36] and the more permissible level widely adopted in most developing countries around the world.

District	Product	Number of Samples	Positive Samples * (%*P*)	AFB_1_ > 2 ng/g		AFT > 4 ng/g		AFB_1_ > 8 ng/g		AFT > 20 ng/g	
				N	%	N	%	N	%	N	%
Bahawalpur	Dates	5	4 (80)	0	0	0	0	0	0	0	0
	Pistachios	5	4 (80)	4	80	03	60	0	0	0	0
	Walnuts	5	5 (100)	4	80	04	80	0	0	0	0
Dera Ghazi Khan	Dates	5	5 (100)	4	80	04	80	0	0	0	0
	Pistachios	5	5 (100)	5	100	05	100	0	0	0	0
	Walnuts	5	5 (100)	5	100	04	80	0	0	0	0
Multan	Dates	5	3 (60)	0	0	0	0	0	0	0	0
	Pistachios	5	4 (80)	4	80	03	60	0	0	0	0
	Walnuts	5	4 (80)	3	60	04	80	3	60	2	40
Rahim Yar Khan	Dates	5	3 (60)	1	20	01	20	0	0	0	0
	Pistachios	5	5 (100)	3	60	03	60	0	0	0	0
	Walnuts	5	5 (100)	3	60	04	80	3	60	1	20
Overall		60	52 (86.7)	36	60	35	58.3	6	10	3	5

* For AFB_1_ and AFT; N: number of samples exceeding the maximum tolerable limits (MTLs) of 2, 4, 8, or 20 ng/g of aflatoxins; %: percentage of samples exceeding the MTLs. Other abbreviations are as defined in Table 1.

Overall, our results suggest that the aflatoxin contamination levels of dates, walnuts, and pistachios marketed in the four studied districts of Punjab (Pakistan) are not too high compared with those reported for the same products in other countries, which can be in the range of mg/g (Table 3). However, they are high enough (most samples exceeded the EU MLTs) to call Pakistani food safety authorities to take immediate measures to face this growing challenge, which has insidious and severe consequences for public health. Therefore, we attempted in this study to assess the potential health risks associated with the consumption of these products in the four studied districts of Punjab (Pakistan), beyond matching the levels of contamination to MLTs. To this end, we used the analytical data obtained in this study, and the dietary exposure of adult male and female consumers as determined by a cross-sectional survey, to determine the risk on the basis of the widely used JECFA approach [41].

### 3.2. Exposure Assessment and Health Risk Characterization of Aflatoxins

The results of EDI and MoE for AFB_1_ and AFT are summarized in Table 5, showing that the EDI values vary greatly depending on the food product, the type of aflatoxin, and the gender of consumers. For both male and female consumers, the highest exposure to AFB_1_ and AFT was unexpectedly recorded from dates, which were the least contaminated, while the lowest exposure was associated with pistachios. The highest exposures to aflatoxins from dates were related to its higher consumption rate (50 g/person/day) compared with pistachios (1.7 g/person/day) and walnuts (5 g/person/day). Conversely, the lower level of exposure through the consumption of pistachios compared with walnuts was more related to the level of contamination (3.32 ng/g vs. 5.59 ng/g). Table 2 also shows that, irrespective of the food item, male consumers were more exposed to aflatoxin intake than female consumers due to the higher consumption rate of males, who are, therefore, at higher risk of developing aflatoxin-related health conditions. Nevertheless, other parameters, such as the pharmacodynamic behavior, the detoxifying efficaciousness of the liver, the age, the general health status, and the socioeconomic status, not considered herein but known to play key roles in the individual’s susceptibility to aflatoxins, may modulate the actual risk status [69]. The average EDI through the consumption of dates, pistachios, and walnuts varied from 0.06 ng/kg bw/day to 0.98 ng/kg bw/day for AFB_1_, and from 0.13 ng/kg bw/day to 2.0 ng/kg bw/day for AFT (Table 5). The maximum exposure to aflatoxins (2.0 ng/kg bw/day) was observed for the male consumers of dates, while the minimum exposure to aflatoxins (0.13 ng/kg bw/day) was recorded for the female consumers of pistachios (Table 5). The exposure data obtained in this study were higher than those reported elsewhere for the same products, or for other nuts and dried fruits. For example, a lower EDI value of 0.29 ng/kg bw/day of AFT from the consumption of dates was reported in Tunisia [65]. Even lower values were reported for Iran, where the exposure to AFB_1_ from the consumption of pistachios and dates was estimated to be 0.013 ng/kg bw/day [70] and 0.12 ng/kg bw/day [28], respectively. The lowest recorded EDI values for AFT worldwide were for countries of the EU, which varied at the upper bound between 0.001 ng/kg bw/day and 0.05 ng/kg bw/day for pistachios and between 0.037 and 0.130 for diverse dried fruits, depending on the country and the GEMS/Food consumption cluster diet [18].

As there are no safe levels/thresholds for the genotoxic and carcinogenic effects of aflatoxins to define reliable benchmarks, such as tolerable daily intake (TDI) to help establish regulatory MTLs, the “As Low As Reasonably Achievable” approach was usually recommended to manage the risk posed by these hazards. Nevertheless, tentative values of 1 ng/kg bw/day and 0.4 ng/kg bw/day have been suggested by Kuiper-Goodman [71] as the provisional maximum tolerable daily intake (PMTDI) of AFB_1_ for hepatitis virus B (HB)-free individuals and HB-carrier individuals, respectively. Accordingly, our results show that none of the analyzed samples reached the PMTDI of 1 ng/kg bw/day for AFB_1_; yet, close values of 0.98 ng/kg bw/day and 0.96 ng/kg bw/day for male and female consumers, respectively, arising from the consumption of dates were recorded. Conversely, the PMTDI of 0.4 ng/kg bw/day was exceeded for both male and female consumers through the consumption of dates, and for males only through the consumption of walnuts (Table 5). Taking into account the fact that the prevalence of HB carriers (HBsAg+) in Pakistan varies from 2.46 to 8.06% [72], the consumption of each of these products would expose the general population of Punjab (Pakistan) to health risks, which should thus be managed by considering the low PMTDI of 0.4 ng/kg bw/day. Contrary to the carcinogenicity of aflatoxins, safe levels for other related adverse health effects, e.g., immunotoxicity, may be established. A recent study demonstrated that Tolerable Daily Intake (TDI) can be determined for the immunosuppressive endpoint of AFB_1_ by using a benchmark approach that was suggested to be suitable for all non-carcinogenic toxicological effects of aflatoxins [44]. According to the latter study, two TDI values (0.017 ng/kg bw/day and 0.082 ng/kg bw/day) were determined, depending on the dose–response curves used to estimate the benchmark dose, which was derived from two different studies conducted on mice strains with inherently different sensitivities to aflatoxins. Considering the TDI value of 0.017 ng/kg bw/day, the consumption of each of the studied food items expose both female and male consumers to a high risk of non-carcinogenic aflatoxin-related intoxications. The risk is higher when the overall aflatoxin intake from consumption of the three products is considered (Table 5).

The potential carcinogenicity arising from the consumption of dates, pistachios, and walnuts individually or in combination was estimated through MoE determinations using average dietary exposure to AFB_1_ and AFT (Table 5). The MoE values obtained (84.83–2857.13) were much lower than the established safe value of 10,000, indicating that the risk for primary liver cancer, e.g., hepatocellular carcinoma (HCC), development among consumers is rather high. The maximum mean MoE value was 2857.13, obtained for the female consumers of pistachios, while the minimum mean MoE was recorded for the male consumers of dates (84.84). For male consumers, the average MoE values by food item varied from 174.14 to 2088.67 for AFB_1_ and from 84.84 to 972.56 for AFT. For the female group of consumers, the average MoE varied from 176.52 to 2857.13 for AFB_1_ and from 86.01 to 1330.39 for AFT. Taken together, the MoE values from the consumption of these three products were significantly lower and varied from 82.13 to 186.81 (Table 5). These results, along with those of the exposure, discussed above, provide evidence that the consumption of dates, walnuts, and pistachios, taken individually or together, pose health risks for the carcinogenic and non-carcinogenic toxic effects of aflatoxins to the surveyed Pakistani population (Table 5). The highest risks were associated with dates, despite their lower levels of contamination compared with pistachios and walnuts due to the higher consumption rate. Limited data regarding the aflatoxin MoE of dates, pistachios, and walnuts are available in the literature. The average MoE values of AFB_1_ obtained in the present study were comparable to those reported for dried dates from Iran [28] and cashew nuts from Nigeria (1000) [73], which also expose Nigerian and Iranian consumers to high risk. However, significantly lower MoE values were found in our date samples (174.14–176.5) than those of Iran (1417), indicating that their consumption exposes consumers in Punjab (Pakistan) to higher health risks. In contrast, the MoE values recorded in our study for nuts were higher than the drastically low MoE of Nigerian groundnuts (6.10) [73]. The prognosis may be worse with the anticipated increase in the consumption of dried fruits and nuts, as per the dietary guidelines and recommendations of different competent organizations intending to boost their consumption rates to 112–150 g/day [12] and 30–42.5 g/day [10,11], respectively.

### 3.3. Effect of Storage Conditions on the Levels of Aflatoxins in Selected Dry Fruits

Figure 1a,b summarizes the results of the concentrations of AFB_1_ and AFT in dates, pistachios, and walnuts at 0, 60, and 90 days of storage under different conditions. At zero days of storage, all the analyzed samples were contaminated with aflatoxins to different, yet not significantly so (*p* > 0.05), extents. However, the levels of aflatoxins in each food item varied significantly (*p* < 0.05), starting from day 60 of storage. Differences recorded between the levels of aflatoxins among the samples under different storage conditions, i.e., open-air storage, hermetic storage in glass jars and cold storage, were also statistically significant (*p* < 0.05). The efficiency of storage conditions in controlling the levels of aflatoxins at the end of each storage period was in the order of cold storage > hermetic storage in glass jars > open-air storage. The aflatoxin concentrations of selected dry fruits stored for 60 days (range 1.27–15.6 ng/g) and 90 days (range 2.0–25.42 ng/kg) were significantly (*p* < 0.05) higher than their counterparts at day zero of storage (range 0.7–3.92 ng/g). Over 90 days of storage, the levels of aflatoxins increased by a factor of 4.0, 2.4, and 1.2 under conditions of open-air, hermetic sealing, and refrigeration at 4 °C (cold storage), respectively. The levels of aflatoxins in all samples were below the EU regulatory MTLs of AFB_1_ (2 ng/g) and AFT (4 ng/g) at day zero of storage. However, at day 90, the aflatoxin concentrations exceeded the European MTLs for both AFB_1_ (Figure 1a) and AFT (Figure 1b) in 56.8% and 76.5% samples, respectively.

In Pakistan, the bulk open-air storage of nuts and dried fruits is a common practice among local retailers, which may contribute to increased aflatoxin contamination reaching unsafe levels before sale. Therefore, appropriate storage conditions are mandatory to prevent or curtail undue increases in aflatoxin levels during storage. Moisture, temperature, and relative humidity are the main environmental parameters to control in order to minimize aflatoxin accumulation during storage [74]. The hermetic storage studied herein significantly reduced (*p* < 0.05) the increase in the levels of both AFB_1_ (Figure 1a) and AFT (Figure 1b) compared with samples stored in open air and under cold storage. The efficacy of hermetic storage compared to open-air storage in preventing or curbing the increase in aflatoxin levels during storage has been reported for other food products, such as peanuts [75], chili pepper [46], and maize [76].

The sealed, waterproofed structure used in hermetic storage creates an internal modified atmosphere rich in carbon dioxide and poor in oxygen, due to the respiration of the biotic components of the stored food product. Such conditions are unfavorable to the growth and aflatoxin production of strictly aerobic aflatoxigenic molds. Hermetic storage was demonstrated to reduce or deplete oxygen in the containers used to store peanuts [75]. The results of the effect of the periodic opening of hermetic glass jars during storage showed no significant impact on the levels of aflatoxins in any one of the stored items (*p* < 0.05). This would allow sellers to occasionally open the jars as required for retail vending, while avoiding excessive accumulations of aflatoxins. Consistent with our findings, Walker et al. [77] found no significant effect of the frequent opening of hermetic bags on the increase in the levels of AFT compared with the control (samples opened only at the end of storage). On the contrary, the weekly opening of maize stored in hermetic bags resulted in increased number of molds and aflatoxin production. This may be due to the higher susceptibility of maize to aflatoxin-producing molds and to the higher moisture content of maize, which exceeded 15%.

Temperature represents a major abiotic factor that affects fungal growth and aflatoxin production in dry fruits during pre- and post-harvest practices. While 16–25 °C is the intermediate ambient temperature range typically used by both producers and consumers to store dried fruits and nuts, refrigeration (4 – 5 °C) is the preferred temperature for household and industrial stores [78]. Aflatoxigenic mold can produce aflatoxins in a wide range of temperatures, spanning from 15 to 35 °C depending on the type of aflatoxins [13]. In the present study, the levels of aflatoxins in the samples stored at 4 °C did not exceed the permissible limit, even after 90 days storage. These results are consistent with those of Alsuhaibani [79], who reported that the levels of aflatoxins in the samples of cashew nuts, pistachios, and walnuts stored at 4 °C for 6 months remained below the regulatory MTL of 15 ng/g for AFT in force in Australian/New Zealand and Iran. Different studies have shown that the main aflatoxin-producing *Aspergillus* species do not grow at temperature below 10 °C [78,80,81].

The findings of the present study suggest that the cold storage of nuts and dates is the most effective means to prevent the excessive accumulation of aflatoxins, followed by hermetic storage, while open air storage is the least effective, as it favors aflatoxigenic mold growth and aflatoxin production. However, hermetic storage would be the most highly recommended, as it is technically and economically more feasible than cold storage, which is unusual for such food products and requires costly refrigeration equipment and maintenance.

## 4. Conclusions and Recommendations

Apart from previous studies surveying the extent of the contamination of dry fruits and nuts with aflatoxins in the northern part of Punjab, Pakistan, this is the first study, to our knowledge, in which the health risks associated with the aflatoxin contamination of dates, pistachios, and walnuts marketed in Pakistan were investigated. The effect of storage conditions on the contamination progress of aflatoxins was also studied for the first time. The findings of the study reveal that an average of 86.7% of samples of dates, walnuts, and pistachios collected from the four districts of South Punjab (Pakistan) were contaminated with aflatoxins, with 60% and 58.3% of the samples exceeded the most stringent EU MTL for AFT and AFB_1_, respectively. Nonetheless, these figures were reduced to 10% and 5% when the relaxed and most widely used MTLs of 8 ng/g (AFB_1_) and 20 ng/g (AFT) were considered. Our results are generally concordant with those of two previous studies conducted in the northern part of Punjab, Pakistan, suggesting that any anticipated official measures to curb the contamination levels of dry fruits and nuts would apply to the whole province. The analytical and consumption data indicate that the highest exposure was associated with date consumption, and the lowest exposure was contributed by pistachios. Male consumers were found to be slightly more exposed to aflatoxins via the consumption of the studied food items than females. Although the concentrations of aflatoxins in most samples were below the widely used regulatory limits of 8 ng/g (AFB_1_) and 20 ng/g (AFT), the risk assessment study suggests that the South Punjab, Pakistan, population is at high risk from the consumption of these products. Indeed, the MoE values for each of the items studied herein were far below the safe limit of <10,000, indicating that the risk is of significant concern to both male and female consumers. This would be of greater concern given the global trend of the increased consumption of dried fruits and nuts, as strongly recommended by international regional and national health and nutrition organizations. The storage conditions of dates and nuts from harvest to consumption are a major parameter that governs aflatoxin accumulation, and any efforts to control the aflatoxin contamination of dry fruits and nuts would be in vain if they are stored under faulty conditions. Of the three storage conditions studied herein, cold storage was found to be the most effective, followed by storage in air-tight containers. Open-air storage, which is presently most commonly used in local markets of Pakistan, was the least effective. The prevalence, levels of contaminations, and risk assessment data obtained in this study invoke the food safety regulatory authorities of Pakistan to establish, monitor, and implement regulatory MTLs for aflatoxins in dried fruits and nuts, so as to ensure the safety of these important diet components in the country. This appears to be especially crucial for dates, whose consumption was associated with the highest health risks owing to their high consumption rate in Pakistan. Such studies should be encouraged, as they may feed the ongoing debate between the Codex Alimentarius Commission (CAC) and the European Food Safety Authority (EFSA) on the advisability of relaxing the EU regulations of AFB_1_ and AFT in dry fruits and nuts. This is especially critical because, on one hand, the international community is encouraged to consume more and more of these products for their taste and health benefits, but on the other hand, their higher consumption will inevitably increase the exposure to aflatoxins. Therefore, the increased consumption of dry fruits and nuts must be paralleled by the development of efficient means to reduce (via prevention or detoxification) their contamination with aflatoxins worldwide, which may open wider the currently reluctant market of industrialized countries to these agricultural products and their derivatives, sourced from developing countries. This would eventually economically benefit both developing and developed countries, while supplying the global market with high-quality and healthy foods.

## Figures and Tables

**Figure 1 ijerph-19-03404-f001:**
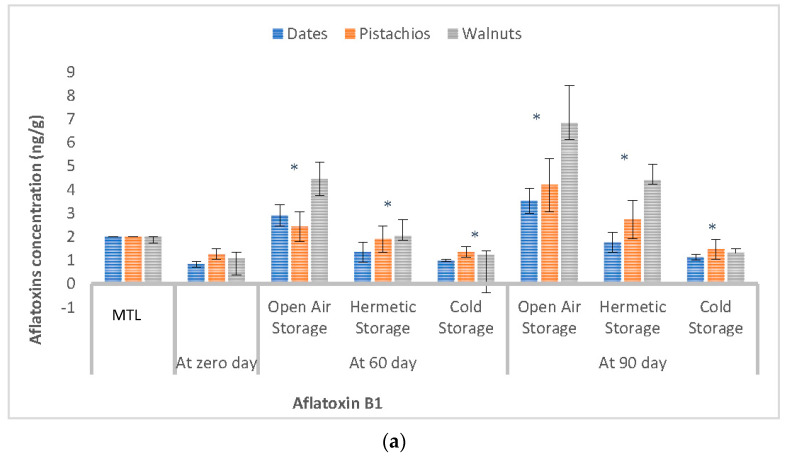
Concentrations (ng/g) of aflatoxin B_1_ (AFB_1_) (**a**) and total aflatoxins (AFT) (**b**) in dates, pistachios, and walnuts at 0 (zero), 60, and 90 days of storage under different conditions as compared with the statutory maximum tolerable levels (MTLs) of 2 ng/g for AFB_1_ (**a**) and 4 ng/g for AFT (**b**) set by the European Union. An asterix mark (*) indicates statistically (<0.05) different concentrations from the control samples.

**Table 1 ijerph-19-03404-t001:** Validation parameters for the HPLC-fluorescence technique used for aflatoxin quantitation in dates, walnuts, and pistachios.

Aflatoxin	% Recovery (±RSD) *			LOD (ng/g)	LOQ (ng/g)
	Dates	Walnuts	Pistachios		
AFB_1_	92.6 (±7.6)	95.7 (±5.2)	96.4 (±5.5)	0.07	0.21
AFB_2_	94.8(±6.6)	93.9 (±3.7)	91.5 (±6.7)	0.03	0.09
AFG_1_	89.7 (±8.4)	86.6 (±7.1)	88.7 (±5.4)	0.07	0.21
AFG_2_	87.8 (±5.8)	85.7 (±4.5)	84.4 (±4.9)	0.03	0.09

* Relative standard deviation (RSD)—AFB_1_: aflatoxin B_1_; AFB_2_: aflatoxin B_2_; AFG_1_: aflatoxin G_1_; AFG_2_: aflatoxin G_2._

**Table 2 ijerph-19-03404-t002:** Prevalence and concentrations of aflatoxins (ng/g) in dates, pistachios, and walnuts collected from four districts of South Punjab, Pakistan.

Punjabi Districts	Product	AFB_1_ *			AFB_2_ *			AFG_1_ *			AFG_2_ *			AFT *		
		Max	Mean (± SD)	%*P* **	Max	Mean (± SD)	%*P*	Max	Mean (± SD)	%*P*	Max	Mean (± SD)	%*P*	Max	Mean (± SD)	%*P*
Bahawalpur	Dates	1.45	0.81 (±0.61) ^a,g^	80	0.85	0.20 (±0.37) ^d^	40	2.03	0.79 (±0.83) ^g^	80	0.18	0.06 (±0.08) ^j^	40	3.32	1.87 (±1.35) ^l^	80
	Pistachios	5.67	2.97 (±2.06) ^b^	80	4.85	1.59 (±2.27) ^e,k^	40	2.98	1.25 (±1.36) ^h,j^	60	2.45	1.43 (±1.01) ^k^	80	13.8	7.25 (±5.65) ^m^	80
	Walnuts	7.62	5.31 (±2.95) ^c^	100	4.52	2.81 (±1.89) ^f^	80	4.59	2.32 (±1.71) ^i^	100	3.65	1.97 (±1.57) ^l,h^	80	19.67	12.40 (±7.80) ^n^	100
Dera Ghazi Khan	Dates	3.98	2.83 (±0.88) ^a^	100	0.92	0.39 (±0.46) ^d,j^	60	1.98	1.46 (±0.31) ^g^	100	1.11	0.44 (±0.51) ^j^	60	7.2	5.13 (±1.40) ^o^	100
	Pistachios	6.73	4.05 (±1.70) ^b^	100	1.24	0.49 (±0.67) ^e^	40	33.76	2.06 (±1.06) ^h,k^	100	2.45	1.29 (±0.89) ^k^	80	13.39	7.90 (±3.16) ^m^	100
	Walnuts	6.98	4.44 (±1.82) ^c^	100	3.78	2.35 (±1.61) ^f,i^	80	55.12	2.96 (±1.46) ^i^		3.44	1.68 (±1.23) ^l^	80	17.85	11.44 (±5.40) ^n^	100
Multan	Dates	1.12	0.64 (±0.59) ^a,g^	60	0.85	0.17 (±0.38) ^d,i^	20	1.09	0.49 (±0.53) ^g^	60	0.84	0.17 (±0.38) ^j^	20	3.02	1.47 (±1.39) ^l^	60
	Pistachios	6.01	3.52 (±2.51) ^b^	80	4.85	1.59 (±2.27) ^e^	40	3.02	1.17 (±1.21) ^h,k^	80	3.15	1.03 (±1.47) ^k^	40	13.72	7.32 (±5.74) ^m^	80
	Walnuts	9.99	6.12 (±4.81) ^c^	80	4.52	2.81 (±1.89) ^f,h^	80	5.63	2.73 (±2.23) ^i^	80	4.63	2.06 (±1.73) ^l^	80	24.2	13.73 (±10.02) ^n^	80
Rahim Yar Khan	Dates	4.34	1.04 (±1.86) ^a^	60	1.43	0.36 (±0.61) ^d^	60	2.09	0.52 (±0.89) ^g^	60	1.76	0.52 (±0.78) ^j,g^	40	9.62	2.44 (±4.08) ^l^	60
	Pistachios	4.98	2.72 (±1.33) ^b^	100	2.12	1.04 (±0.76) ^e,k^	80	3.24	1.23 (±1.29) ^h^	80	3.15	1.03 (±1.47) ^k^	40	10.34	6.03 (±3.17) ^m^	100
	Walnuts	9.99	6.22 (±4.67) ^c^	100	4.52	2.81 (±1.89) ^f,i^	80	4.87	2.73 (±1.95) ^i^	80	3.45	1.70 (±1.29) ^l^	80	22.26	13.47 (±9.59) ^n^	100
Overall ***		9.99	3.39 (±2.96)	86.7	4.85	1.39 (±1.68)	58.3	5.63	1.63 (±1.48)	81.7	4.63	1.12 (±1.23)	60	24.2	7.54 (±6.68)	86.7

* Minimum values were <LOD for each aflatoxin type; ** prevalence (percentage of positive samples); *** average concentration/prevalence of toxins in all studied items from the four studied districts of South Punjab, Pakistan; AFT: total aflatoxins (sum of AFB_1_, AFB_2_, AFG_1_, and AFG_2_); ^a–n^: mean values sharing the same superscript are not statistically different (*p* > 0.05). Other abbreviations are as defined in Table 1. Note: the minimum values for all aflatoxins are below the limit of detection (LOD); the LOD and the limit of quantification (LOQ) values are presented in Table 1.

**Table 5 ijerph-19-03404-t005:** The average daily consumption of dates, pistachios, and walnuts, calculated estimated daily intake (EDI) and the margin of exposure (MoE) for AFB_1_ and AFT.

Product	Mean Concentration (ng/g) * ±SD		Average CR (g/Person/Day) ± SD		EDI (ng/kg bw/Day)				MoE			
					AFB1		AFT		AFB1		AFT	
	AFB1	TAF	Male	Female	Male	Female	Male	Female	Male	Female	Male	Female
Dates	1.33 ± 1.36	2.73 ± 2.62	50 ± 6.19	40 ± 7.59	0.98	0.96	2.00	1.98	174.14	176.5	84.83	85.99
Pistachios	3.32 ± 1.87	7.13 ± 4.29	1.67 ± 0.32	0.99 ± 0.46	0.08	0.06	0.17	0.13	2088.7	2857.1	972.56	1330.39
Walnuts	5.52 ± 3.54	12.76 ± 7.76	4.56 ± 0.72	3.43 ± 0.77	0.37	0.34	0.85	0.79	460.1	496.0	199.02	214.56
Overall	3.39 ± 2.96	7.54 ± 6.65	18.7 ± 22.4	14.8 ± 18.4	0.93	0.91	2.07	2.02	182.80	186.81	82.13	84.16

EDI: estimated daily intake; CR: consumption rate; MoE: margin of exposure; AFT: total aflatoxins (sum of AFB_1_, AFB_2_, AFG_1_, and AFG_2_). Other abbreviations are as defined in Table 1. * Mean concentrations of determinations by food item collected from the four districts. Note: the average wights (±SD) of the surveyed females and males were 55.24 ± 5.90 and 68.12 ± 4.44 kg, respectively.

## Data Availability

Data archived by the author Amir Ismail.

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
