# Peer review of "Prevalence of Aflatoxins in Selected Dry Fruits, Impact of Storage Conditions on Contamination Levels and Associated Health Risks on Pakistani Consumers"

_ijerph, 2022, doi:10.3390/ijerph19063404_

Round 1
Reviewer 1 Report
The authors consider their findings on the natural occurrence of aflatoxins in dry fruits (dates, pistachios and walnuts) from Pakistan represent an advance in knowledge of the potential health risk.
I have a few comments that might be useful for the authors.
Page 3, in line 23 of 2. Materials and Methods 21, 2.1. Sampling:
“A total of 60 samples comprised of 5 samples of each of dates, pistachios, and walnuts”
There is a lack of clarity in this sentence. It should be re-written.
Page 3, lines 42-44: "the solution to the final volume of 1.95 mL vortex for 1 min. The lower layer containing aflatoxins 43 was separated using syringe filters (0.45μm) before proceeding to chromatographic analysis."
It is not clear and should be re-written. Moreover, the authors should consider to describe the utility of the syringe filters (0.45μm) to separate the lower layer.
Page 5, in line 22 of 2.9. Effect of Storage Conditions on the Levels of Aflatoxins 19
“for visual fungal contamination“
Do you have any citations backing this statement up? If so, please include them.
Page 6, lines 15-17:
“wherein the LOD and LOQ values obtained in this study fit well [48], which still includes the values found in our study. “
There is a lack of clarity in this sentence. It should be re-written.
Page 6, lines 35-36:
“average concentrations of 12.76 ng/g, 7.13 ng/g, and 2.73 ng/g for AFT and 5.59 ng/g, 3.32 ng/g, and 1.33 ng/g (Table2).”
There is a lack of clarity and info in this sentence. “Table 2” should be “Table 4“. Moreover, “5.59 ng/g, 3.32 ng/g, and 1.33 ng/g FOR ?” It should be re-written.
Table 2
“mean values having different letters within the columns are statistically significant (P < 0.05).”
The authors should consider to describe the meaning of different letters. It is not clear and should be re-written.
Page 12, Lines 1-10: what is the idea the authors are trying to convey? It is not clear and should be re-written.
Reviewer 2 Report
This manuscript focuses on health risks tied to aflatoxins contamination of dry fruits. Despite tons of papers have been already published on this topic since the discovery of the aflatoxin effects and conditions favouring their production, the authors purpose a study combining the aflatoxin quantification (from different nuts/fruits stored in different conditions) to an epidemiological investigation. This last aimed to quantify the risk for the resident population.
This paper could also open to the global scale problem due to the increasing trend in nuts /dry food consumption in industrialized countries (that are rediscovering healthy foods) -as mentioned in the introduction - and winking to the not neglectable economic aspect. A high-quality product that combines taste and safety can be a resource to be exported and economic wealth for each country. Maybe some highlights on conclusions/discussion could be useful to support future investigations.
The paper is well written and requires a few minor improvements before its publication.
Abstract - it’s better to shortly report the aims (all of them) and the research design before to report results. In particular “. The estimated daily intake (EDI) and margin of exposure (MoE) of aflatoxins through the consumption of the studied dry fruits ranged from 0.06 ng/kg bw/day to 2.0 34 ng/kg bw/day and from 84.84 to 2857.13” has been introduced without mentioning the epidemiologic investigation performed.
Keywords: if you want to increase your visibility you must work on keywords…they are too general and some of them are already present in the title and so already indexed.
Results can be improved in their form. The reader should have in single sight the possibility to catch the salient data even in a table. Tables and figures should be considered as standalone material so they must contain all information to correct reading/understanding. E.g., Table 1 doesn’t report in the caption LOD LOQ meaning.
Below are some additional notes.
Minor notes
L26 (dates, pistachios, and walnuts) it’s a repeat of the opening sentence please modify it to avoid redundancies.
L46-49 are grey colour written
L 53 669.86thousand missing space.
Pg 3 L28 until needed for analysis…wordy, please change to “until analysis”
L34 16 mL please remove, is an exceeding information
L39 Lof missing space
L41 placed to react change to left to react
L42 was added in the vials to makeup the solution to the final volume of 1.95 mL vortex for 1 min. this sentence doesn’t sound good. Please change to: was added up to the final volume of 1.95 mL and vortexed for 1 min
L50 1mL/min missing space
Page 4
L1 t 365nm and 440nm missing space
Table 1 should be ameliorated in terms of text distribution (see pistachios column). The column width should be more regular as the numbers are shown in a single line. More, there are AFB1 and date’s results in bold and relative cells underlined. Please check
Table 3. check the width of the rows, that ones relative to dates are larger than others.
Next time pay attention to line numbering because it restarted on each page, this fact delays reviewing procedure
Page 12 L41-44 are in bold, from L41 to the following page are in Palatino 9 instead 10.
Table 4 please be consistent writing MoE/MOE , enlarge also the product’s column
Reviewer 3 Report
The manuscript deals with the important problem of the presence of aflatoxins in dried fruit and nuts. The authors undertook studies on the occurrence of aflatoxins, consumer exposure, and the influence of storage conditions on their level. The subject matter described is topical and fits in with the important trends in food control.
The article is written correctly, I recommend its approval if the authors correct and refer to the following issues:
- Please pay attention to the formatting of the entire article in accordance with the requirements of the journal. The font is sometimes mixed up, discontinuous line numbering made the review difficult.
- The discussion of the results is a bit too detailed, especially the part concerning the comparison of the levels of aflatoxins in nuts and dried fruits, it is a kind of "enumeration", I recommend that you improve the discussion for a more accessible one.
- page 2, lines 16/17 in vitro, in vivo: please apply the italics
- page 2, lines 51, 53, 55... is the notation e.g. 1022.03 thousand MTs correct? Please check it.
- page 4, line 19 and page 10 line 17 - place cite the papers of Golge et al. and Ozer et al. properly (missing the numbers)
- Point 2.4, please specify the aflatoxin standards used
- page 7 line 13, the word "of" is missing in the second bracket
- page 7 line 22, please correct "AFB1butsomewhathigher"
- page 7, line 37/38 is the unit μg/kg correct? It should be ng/kg?
- Table 3, columns 3 and 4, please write the numbers as "5" instead of "05" etc.
- Figure 1 - Can you mark the statistically significant differences in relation to the control (day 0), e.g. with *?
Reviewer 4 Report
The subject of the article is very interesting. The problem of molds, which produce aflatoxins that can accumulate in dried fruit, nuts, and foods with low water activity is quite common.
The Authors did not avoid errors in the preparation of the test results.
Here are some of my comments:
- Most comments are related to the literature on the subject - mainly the method of citing and references. Eg page 10 line 17 "Ozer, et al. [2012]" rather it should be "Ozer et al. [62]".
References are completely free to write: some entries have hyperlinks inserted; items number 40 and 55 are missing data from the journal; position number 57: the surname "WANG" instead of "Wang"; again, in position 60, the title was written in capital letters. It is also not appropriate to refer the reader in the manner written on page 2, line 14: "for review, see [7, 8]." - Point 2.1 Sampling: I wonder if the bulk containers from which the fruit samples were collected were available in the consumer store? Did the Authors take samples from these containers immediately after their first opening? This is of paramount importance for potential mold spore contamination and may influence subsequent results. maybe a better solution would be to get these samples directly from the manufacturers?
- Selection of dried fruit sampling sites. The Authors write about the selection of 4 districts in Punjab. What was the guiding principle when selecting these places? Do they differ in the specificity of the climate, temperature or humidity? Please add a few sentences to the environmental characteristics of these sites, which may have a significant impact on the growth of mycotoxin-producing molds.
- Markings in the raw material. I believe that at least one of the two determinations that should be made on such a material has been omitted. One is the determination of the water activity and the other is the water content. One of them would be enough to indicate additional parameters of the environment in which molds produce mycotoxins.
- Descriptions of determinations in the research methodology: points 2.2, 2.4, 2.6 and 2.7 - no reference to any source. Please complete the information, whether they are my own studies or according to some literature.
- Point 2.9 "Effect of storage ..." - please indicate on what basis such storage conditions as described in the experiment were selected.
- Table 2. It was a bad idea to put the results for AFB1 and Prevalence (%) in one column. It will be more explicit to add another column for this result.
- Page 12, line 20 - in the text the authors refer to table number 2 and the description shows that it should be table number 4. Please check this number.
